# Improvement of Zn (II) and Cd (II) Biosorption by *Priestia megaterium* PRJNA526404 Isolated from Agricultural Waste Water

**DOI:** 10.3390/microorganisms10122510

**Published:** 2022-12-19

**Authors:** Othman M. Alzahrani, Aly E. Abo-Amer, Rehab M. Mohamed

**Affiliations:** 1Department of Biology, Faculty of Science, Taif University, P.O. Box 888, Taif 21974, Saudi Arabia; 2Department of Botany and Microbiology, Faculty of Science, Sohag University, Sohag 82524, Egypt

**Keywords:** *Priestia megaterium*, biomass, zinc, cadmium, biosorption, 16S rRNA, Langmuir and Freundlich isotherm equations

## Abstract

Heavy metals are considered as dangerous pollutants even in relatively low concentrations. Biosorption is an ecofriendly technology that uses microbial biomasses for adsorbing heavy metals from wastewater on their surfaces based on physicochemical pathways. Ten agricultural wastewater samples were collected from different sites in Sohag Governorate, Egypt. One hundred and nineteen zinc and cadmium-resistant bacterial isolates were recovered from the water samples. Interestingly, the isolate R1 was selected as the most resistant to Zn^2+^ and Cd^2+^. This isolate was morphologically and biochemically characterized and identified by sequencing of 16S rRNA gene as *Priestia megaterium*, and then deposited in the GenBank database under the accession number PRJNA526404. Studying the effects of pH and contact time on the biosorption process revealed that the maximum biosorption was achieved within 50 min at pH 7 and 8 for Zn^2+^ and Cd^2+^, respectively, by the living and lyophelized biomass of *Priestia megaterium* PRJNA526404. The preliminary characterization of the main chemical groups present on the cell wall, which are responsible for heavy metal biosorption, was performed by Infrared analysis (IR). Kinetics studies revealed that data were fitted towards the models hypothesized by Langmuir and Freundlich isotherm equations. The maximum capacity values (*q_max_*) for biosorption of zinc and cadmium reached by using living and lyophelized biomass were 196.08; 227.27 and 178.57; 212.777 mg/g, respectively, and it was indicated that lyophilization improved efficiency of the biomass to heavy metals compared to living cells. The results indicated that *Priestia megaterium* PRJNA526404 had good application prospect in cadmium and zinc water remediation.

## 1. Introduction

The environmental pollution has accelerated over the last few decades as a result of the increase in the activities of man such as industrialization, mining, plastic and oils production and application of fertilizers. The increasing of public attention to the pollution of water bodies by metals is due to its toxicity for flora and fauna, and for humanity [1]. Two million tons of waste are discharged to water bodies daily, including industrial wastes, dyes, heavy metals and chemicals, and human and agricultural wastes (pesticides or fertilizers) [2]. The toxicity of heavy metals is due to their mobility in natural water, they are highly persistent and are not degradable in the environment, and they may result in health problems through accumulation in microorganisms, aquatic flora, and fauna, which, in turn, may enter into the human food chain and then directly affect human health [3,4,5]. Among the prevalent elements in the Earth’s crust, zinc is considered as an essential trace element for living organisms. It is a significant component of various cofactors or coenzymes of more than 300 enzymes [6,7]. Although being an essential trace element, zinc can cause harmful health effects even at very low concentration [8,9]. Cadmium ions accumulate in internal body organs as the liver, lungs, pancreas and bones causing apoptosis, osteoporosis and respiratory tract problems [10,11]. It is cytotoxic at low concentrations and could lead to human cancer [12]. Over the last few decades, heavy metals were removed from contaminated wastewaters via many conventional treatment methods. These methods, including ion exchange, chemical precipitation, ultra-filtration, reverse osmosis and electro winning, are commonly used [13]. However, most of these methods are complicated and expensive; they are also ineffective when applied to lower concentrations of metal ions [14]. New ecofriendly, cheaper and more effective technology, based on metal ions, the removal by adsorption is significantly economic, convenient and an easy operation technique. It shows high metal removal efficiency and is applied as a quick method for all types of wastewater treatments [15]. Different materials may be used as metal adsorbents and may be obtained from organic, mineral or biological sources (e.g., agricultural waste, aquatic and terrestrial biomass and other locally available waste materials). It is becoming a popular technique because, in this process, the adsorbent can be reused, and metal recovery is possible [16]. Biosorption can be defined as the ability of biological materials to absorb heavy metals from a solution metabolically mediated or physicochemically [17]. Microorganism processes, such as bioaccumulation and biosorption, have been applied for heavy metal removal from contaminated environments [12,18].

Bacterial biomass has been found to be one of the most important biosorbents used for metal removal among different microbial populations [19]. Several available adsorption isotherms are adopted to correlate adsorption equilibrium in heavy metals’ biosorption. Langmuir’s and Freundlich isotherms are widely used. The parameters affecting sorption, such as pH and contact time, have also been studied [4].

The present study aims to assess and compare between the values of biosorption capacities of Zn^2+^ and Cd^2+^ by untreated (living) and treated (lyophilized) biomass of a heavy metal-resistant bacterial strain recovered from wastewater.

## 2. Materials and Methods

### 2.1. Isolation and Characterization of Bacteria

Ten samples of wastewater were collected from different sites at Sohag Governorate (Egypt), and they were collected in sterile screw–capped bottles and transferred immediately to the laboratory. Bacteria were isolated on nutrient agar; then, their heavy metal resistance was screened using the agar dilution method on a modified Tris Minimal Medium (Sigma) supplemented with different concentrations of Zn^2+^ and Cd^2+^ (100 to 1000 ppm) as ZnSO_4_ (Oxoid) and CdCl_2_ (Oxoid), respectively. The most resistant isolate was selected for further investigations [20]. The isolate was purified, then, tentatively identified on the basis of the classification schemes published in Bergey’s Manual of Systematic Bacteriology [21].

### 2.2. DNA Extraction, Sequencing, and Phylogenetic Analysis

Bacterial genomic DNA of the heaviest metal-resistant isolate was extracted using Insta Genetm Matrix (BIO-RAD). The primers 27F 5′ (AGA GTT TGA TCM TGG CTC AG)3′ and 1492R 5′ (TAC GGY TAC CTT GTT ACG ACT T)3′ were used for the PCR. The reaction of PCR was performed with 20 ng of DNA as the template in a 30 µL reaction mixture by using EF-Taq (SolGent, Daejeon, Korea). Cycling parameters were as follows: an initial step of 98 °C for 2 min, followed by 35 cycles of 95 °C for 5 s, 55 °C for 15 s, and 72 °C for 20 s, and completed with a final step of 72 °C for 1 min. The amplification products were purified with a multiscreen filter plate (Millipore Corp., Bedford, MA, USA).

### 2.3. DNA Sequencing and Phylogenetic Tree

A PRISM Big Dye Terminator v3.1 Cycle sequencing Kit was used for the performance of the Sequencing reaction. Extension products contained by the DNA samples were added to Hi-Di formamide (Applied Biosystems, Foster City, CA, USA). The mixture was incubated for 5 min at 95 °C, followed by 5 min on ice before analysis by a DNA analyzer (ABI Prism 3730XL, Applied Biosystems, Foster City, CA, USA). The primers 518F 5′ (CCA GCA GCC GCG GTA ATA CG)3′ and 800R 5′ (TAC CAG GGT ATC TAA TCC) 3′ were used for sequencing. Sequences were aligned with others retrieved from Gen Bank using ClustalX and manually optimized [22]. 

Positions with species contained a length mutation, and ambiguously aligned regions were not included in the subsequent phylogenetic analysis. Maximum likelihood and maximum parsimony analyses were made using PAUP 4 [23]. One hundred random addition heuristic search replicates, and 1000 bootstrap replicates were performed employing 5 random addition heuristic searches to obtain maximum-parsimony (MP) trees.

Maximum-likelihood (ML) analysis was accomplished using heuristic searches with the tree bisection-reconnection (TBR) rearrangements and random stepwise addition of 100 replicates [24]. Regarding hierarchical likelihood ratio tests (hLRTs), Modeltest 3.7 was used to determine the optimal model of nucleotide substitution for the ML analyses [25]. TrN was selected as the best fit for the 16s rDNA dataset. The phylogenetic trees were visualized using Njplot and were edited via Adobe Illustrator CS6 [26].

### 2.4. Preparation of Priestia Megaterium PRJNA526404 Biomass

Bacteria were cultured in nutrient broth at 37 °C for 24 h, with agitation at 150 rpm. Cells were collected by centrifugation at 10,000 rpm for 10 min using (Eppendorf centrifuge, PLC-012). Suspensions were rinsed with sterile distilled water three times before freeze drying by (Labconco 94814) bench-top lyophilizer [27,28].

### 2.5. FTIR Spectroscopy

Fourier transform infrared spectra (FTIR) were recorded for raw samples, and biomass was loaded with 50 ppm Zn^2+^ and Cd^2+^ on a 470 Shimadzu FTIR spectrometer.

### 2.6. Effect of pH and Time on Biosorption

The influence of the solution pH on the biosorption of Zn^2+^ and Cd^2+^ was investigated for the *Priestia megaterium* PRJNA526404 biomass. Biosorption was conditioned to different pH environments (ranging between pH 2 and pH 10) containing 20 mL of metal solution. Either 0.1M NaOH or 0.1M HCl were added to adjust pH. For all experiments, sodium nitrate (0.1M) was used as a supporting electrolyte. This method was performed according to Seki et al. [29]. Experiments for determination of the kinetics of the biosorption process were carried out using 50 mg L^−1^ of the initial metal concentrations of Zn^2+^ and Cd^2+^ in 20 mL of metal solution.

### 2.7. Adsorption Procedure

The biosorption isotherms of Zn^2+^ and Cd^2+^ ions were achieved at pH 7 and 8, respectively. The biosorption experiments were carried out using 20 mg of the untreated (living) biomass or lyophilized cells along with different concentrations of Zn^2+^ and Cd^2+^ starting from 0 to 300 ppm contained in 20 mL of 0.1 M NaNO_3_ as a supporting electrolyte solution with agitation for 50 min at 200 rpm to attain equilibrium. Experiments were performed at room temperature (30 ± 2 °C). The samples were centrifuged for 5 min at 10,000 rpm; then, the heavy metal concentrations in supernatants were measured by the atomic absorption spectrophotometric (AAS) Model 210 VGP Buck Scientific.

### 2.8. Evaluation of Data

The amount of Zn^2+^ and Cd^2+^ adsorbed by bacterial biomass was calculated from the differences between the initial metal quantity added to the biomass and the remaining metal content in the supernatant by using the following equation:(1)$$q_{e}=\frac{\left(C_{0}-C_{e}\right)V}{m}$$
where *q_e_* is the adsorption quantity obtained at the equilibrium (mg g^−1^), and *C*_0_ and *C_e_* are the initial and equilibrium concentrations (the solution concentration at the end of the sorption process) (mg/L), respectively. *V* is the volume of the solution (L), and *m* is the weight of biomass (g) in grams.

### 2.9. Adsorption Isotherms

Adsorption isotherms represent the equilibrium between an adsorbate on an adsorbant and the adsorbate remaining in the aqueous phase. The data obtained from adsorption of heavy metals were fitted to Langmuir and Freundlich models to better define the biosorption process.

Langmuir isotherm: the Langmuir isotherm (Equation (2)) is used to describe mono-layer adsorption, and it postulates that the binding sites on the biosorbent are homogeneous or have equal affinity for the heavy metal ions [30]:(2)$$\frac{C_{eq}}{q_{eq}}=\frac{1}{q_{max}\;b}+\frac{C_{eq}}{q_{max}}$$
where *q_eq_* represents the capacity for heavy metal mass per unit mass of biomass (mg/g) at equilibrium (solid phase concentration), and *C_eq_* is the concentration of heavy metal remaining in solution (mg/L) at equilibrium (liquid phase concentration), respectively. *q_max_* and *b* are the maximum binding capacity in mg of heavy metal per g (mg/g) of biomass and the Langmuir adsorption binding constant (L/mg), respectively.

Freundlich isotherm: another way for analyzing the adsorption data is to apply the Freundlich Equation (3) [31]:(3)qeq=Kf Ceq1n
where *q_eq_* and *C_eq_* are as the same as previously defined. K_f_ is the Freundlich distribution coefficient (mg/g) and *n* is the adsorption intensity (L/mg).

## 3. Results and Discussion

### 3.1. Isolation and Identification of Bacteria

One hundred and nineteen bacterial isolates were recovered from 10 waste water samples.

All these isolates were investigated for their capability to grow on nutrient agar medium with different concentrations (50–1000 ppm) of Zn^2+^ and Cd^2+^. Only the bacterial isolate R1 was able to survive up to (500–700) ppm Cd^2+^ and Zn^2+^, respectively, and it was selected for more investigations. Based on the microscopic examination and the biochemical activities in (Table 1) according to Bergey’s manual, the isolates were presumptively identified *Priestia megaterium*. About 1500 bp of 16S rRNA gene sequences of the isolate R1 were amplified using the universal primer for the 16S rRNA gene and sequenced. A phylogenetic tree was constructed by aligning 16S rRNA sequences of different bacteria taken from NCIB and the sequence of the isolate from this study (Figure 1). The isolate R1 was identified as *Priestia megaterium* and assigned the accession number PRJNA526404 in the Gene Bank database.

### 3.2. Cell Surface Change by FTIR Analysis

Biosorption of metal ions by the microbial biomass is largely dependent on the physiochemical conditions of the solution and the functional groups present on the active sites of the bacterial cell surface. FT-IR analysis of *Priestia megaterium* PRJNA526404 biomass was carried out to understand the types of functional groups that may be involved in the biosorption process. By comparing the infrared spectra of *Priestia megaterium* PRJNA526404 before and after adsorbing Zn^2+^ and Cd^2+^ (Figure 2), it can be noticed that the location of maximum absorption peak of the hydroxyl group location moved from 3422.1 cm^−1^ to 3415 cm^−1^ and 3414 cm^−1^; the absorption peak at 2939.3 cm^−1^ reflected the stretching vibration of the alkyl in the protein, carbohydrates along with other substances, and it was slightly shifted to 2935 cm^−1^ after adsorption of Zn^2+^ and Cd^2+^. The band at 1654.2 cm^−1^ referring to amide groups has moved to 1649.3 cm^−1^ and 1653.2 cm^−1^. The absorption peaks caused by C=C and methyl groups at 1560.6 and 1392.8 shifted to 1546.1 cm^−1^, 1551.0 cm^−1^ and 1387.0 cm^−1^, 1388.0 cm^−1^. The band localized at 1251.0 cm^−1^ amide II zone (N-H bending vibration and C-N stretching vibration superposition) disappeared after Zn^2+^ adsorption and migrated to 1247.1 cm^−1^ after adsorption of Cd^2+^. Moreover, the new band that appeared at 1155.5 cm^−1^ was attributed to the C-O group after the loading of Cd. Such observations provide evidence that functional groups like hydroxyl, amino, amide, carbonyl, and carboxyl groups are involved in the binding of Zn and Cd on surface of biosorbents. These findings come in agreement with Mosa et al. and Tarekegn et al., who reported that bacteria such as *Bacillus*, *Micrococcus* have great metal biosorption ability due to their high surface-to-volume ratios and potential active chemisorption sites (teichoic acid) on the cell wall [32,33,34]. Similar data have also been reported in the literature [35,36,37,38].

### 3.3. Effect of pH on Biosorption

Figure 3 demonstrates the influence of pH on the biosorption efficiency of Zn and Cd at different pH value range 2–10, and the metal uptake increased by increasing the pH up to a maximum above which the metal uptake decreases as the pH increases. The living and lyophilized biomass achieved an optimal adsorption of Zn^2+^, and Cd^2+^ was at pH 7 and 8, respectively. The pH has a strong influence the chemistry of the metal solution and the activity of functional groups (phosphate, carboxylate, and amino groups) on the cell wall. It also affects the competition of metallic ions for the binding sites [39,40]. Edris et al. reported that cadmium and lead maintained the highest removal biosorption using *C. vulgairs* at a pH value of around 7 [41], but at low pH, adsorption decreases, which may be attributed to the competition between high H^+^ content in the solution and metal ions on the binding sites of biosrobent surface, making metals’ sorption difficult [42]. On the other hand, at high pH values, the solubility of metals is significantly reduced, and metal hydroxides are formed, which collide and thus impede biosorption [43].

### 3.4. Effect of Contact Time on Biosorption

The amount of adsorbed Zn^2+^ and Cd^2+^ (mg/L) plotted against time (min) at the optimum pH is shown in (Figure 4a,b). An increase in biosorption rate is observed over the first 50 min, then the rate tends to be constant. Such a short time required for biosorption is in accordance with the results given by other studies [4,34,44,45,46].

This illustrates that the biosorption consists of two phases: a primary rapid phase that accounts for the major part in the total metal biosorption—this is due to the high initial Zn and Cd concentrations and empty metal binding sites on the cell surfaces—and a second slow phase due to the saturation of metal binding sites [44]. This suggests that most of the metal ions are adsorbed within the first 10 min, which may be attributed to the interaction with functional groups located on the bacterial cell surface, then a secondary phase in which the rate decreases till reaching a constant value of metal concentration after 50 min; this represents the time of equilibrium at which an equilibrium of metal ion concentration is attained. The initial fast metal biosorption rate may be attributed to the surface binding by natural particles, and the following slower adsorption to the interior penetration [47].

### 3.5. Adsorption Isotherms

Zinc and cadmium biosorption performance on *Priestia megaterium* PRJNA526404 biomass was evaluated by the biosorption equilibrium measurements at an initial concentration of 0 up to 300 mg/L for both metals during 50 min and at 7.0 and 8.0 pH values, respectively, as postulated in Figure 5a,b.

The biosorption isotherm curves are used to describe the equilibrium distribution of metal ions between the aqueous and solid phases [48]. The Langmuir model considers the adsorbate adsorption limited to one molecular layer at or before a relative pressure of unity is reached [49]. The Langmuir isotherm model is used to estimate maximum adsorption capacity *q_max_* (mg/g) and adsorption coefficient constant b (L/mg). The Freundlich model assumes a heterogeneous biosorption system with different active sites [50]. The Freundlich isotherm model is used to estimate the adsorption intensity n (L/mg) and the adsorption capacity K_f_ (mg/g). The performance of Zn^2+^ and Cd^2+^ biosorption on *Priestia megaterium* PRJNA526404 cells was accomplished by the biosorption equilibrium measurements at concentrations from 0 up to 300 ppm for both metals after 50 min at pH values of 7.0 and 8.0, respectively. Langmuir and Freundlich constants were calculated from Equations (2) and (3) and through the corresponding plots of the metal biosorption as described in Figure 6 and Figure 7, which are summarized in Table 2 as well. Data revealed that the *q_max_* values obtained for Zn^2+^ and Cd^2+^ by lyophilized cells were 227.27 and 212.777 mg/g, respectively, which were higher than *q_max_* values obtained using living biomass—196.08 and 178.57 mg/g, respectively. These values are higher than those obtained by Li et al., which were 68.31 and 62.85 mg/g by *Pseudomonas putida* (PAB-0031) biomass, respectively [51]. Mohapatra et al. demonstrated that the maximum uptake of Pb^2+^ (216.75 and 207.4 mg/g) was obtained with live and dead biomass of *Bacillus xiamenensis* PbRPSD202 [52]. These values match well with data previously reported in the literature; see Table 3.

The values of Freundlich parameters summarized in Table 2 show that adsorption capacities K_f_ for Zn^2+^ and Cd^2+^ using living and lyophilized biomass were 7.36, 13.93 and 6.49, 9.82 mg/g, respectively. Summaries of linear regression data (r^2^) for Langmuir and Freundlich isotherms for Zn^2+^ and Cd^2+^ biosorption described in (Table 2) show that (r^2^) values obtained for Zn^2+^ and Cd^2+^ from Freundlich isotherm models were higher than those obtained by a Langmuir isotherm. In this manner, the Freundlich isotherm evidently fits better with the equilibrium of biosorption of Zn^2+^ and Cd^2+^ than the Langmuir isotherm model. This comes in agreement with Khan et al., who reported that K_f_ values for Cd^2+^ were 10.76 and 12.13 mg/g adsorbed by living and dead biomass of *Salmonella enterica* 43C, respectively [56].

The biosorption process achieved by *Priestia megaterium* PRJNA526404 is diagrammatically summarized in Figure 8.

### 3.6. Effect of Pretreatment

In general, passive uptake of heavy metals using dead or lyophilized cells is more efficient than active uptake by untreated living or untreated cells. This is believed to be due to the fact that passive uptake is independent of energy, unlike active uptake, which is metabolically dependent, requires energy for metal transfer across the cell wall into the cytoplasm and, additionally, the presence of competition by the H^+^ produced by viable cells; such competition is absent during passive uptake.

The high adsorption capacity of the cells of *Priestia megaterium* strain could be referred to increasing the surface area in addition to the exposure of intracellular binding sites, which are caused by freeze drying. These findings come in accordance with the results reported by other studies [4,34,37,42,43,57,58].

## 4. Conclusions

Based on the laboratory experiments, one can conclude the following: The bacterial isolate *Priestia megaterium* PRJNA526404 recovered from sewage water is highly resistant to Zn^2+^ and Cd^2+^ ions. The freeze-dried biomass of this isolate has proved to be a promising biosorbent for zinc and cadmium removal from aqueous solutions. The maximum biosorption of Zn^2+^ and Cd^2+^ could be achieved after 50 min at room temperature and at 7.0 and 8.0 pH values for Zn^2+^ and Cd^2+^, respectively. The adsorption data fitted both Langmuir and Freundlich models well for metal ions in the investigated concentration range. The maximum adsorption uptake (*q_max_*) of zinc and cadmium calculated from the Langmuir equation for biosorption by living and lyophilized biomass is 196.08, 227.27 and 178.57, 212.777 mg/g, respectively. More investigations are required for studying the effect of different parameters on the biosorption process and to clear the mechanisms of action to ensure complete bioremediation of zinc and cadmium metals.

## Figures and Tables

**Figure 1 microorganisms-10-02510-f001:**
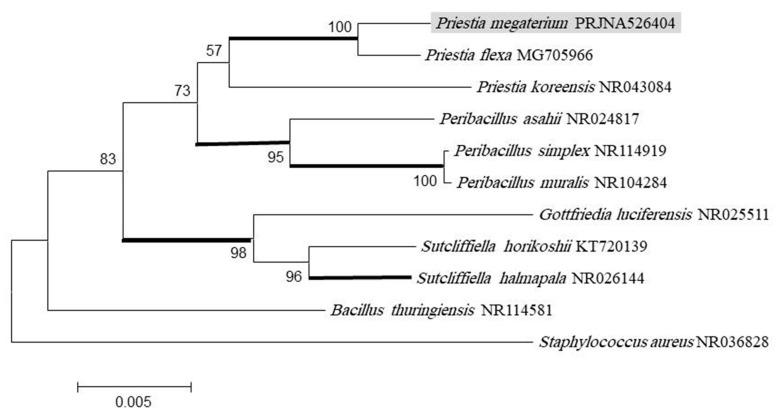
Phylogenetic relationships of *Priestia megaterium* and four of the same species retrived from Gen Bank and related species and genera in the family Bacillaceae on the nucleotide sequences of LSU rDNA. The strain used in the present study is highlighted. The maximum likelihood tree (ML) was constructed in MEGA6. The numbers indicate pp values ≥ 95 % (in bold), MP, ML and MP bootstrap values ≥ 70.

**Figure 2 microorganisms-10-02510-f002:**
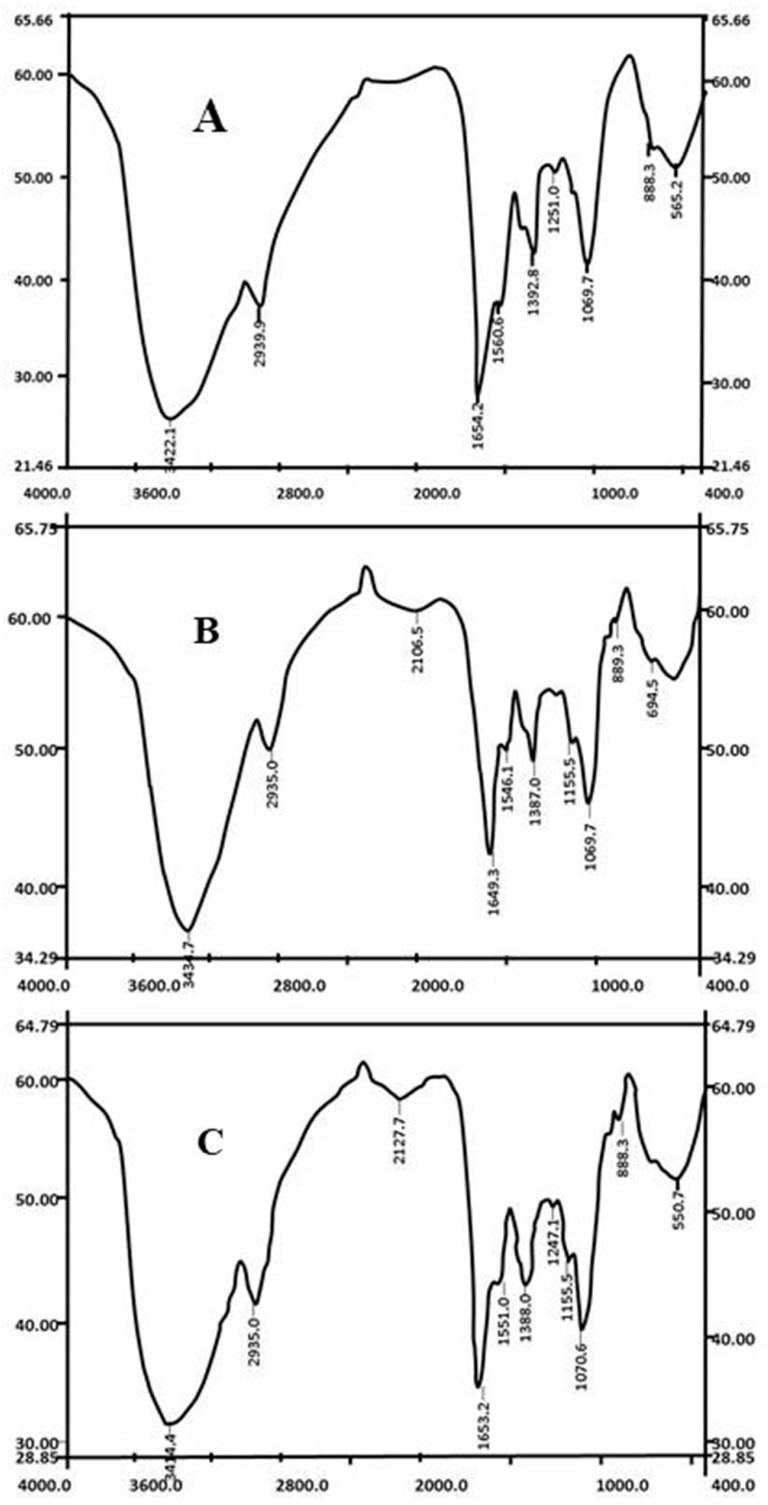
IR spectrogram of *Priestia megaterium* PRJNA526404 before abosorbing (**A**) and after the absorbing of Zn^2+^ (**B**) and Cd^2+^ (**C**).

**Figure 3 microorganisms-10-02510-f003:**
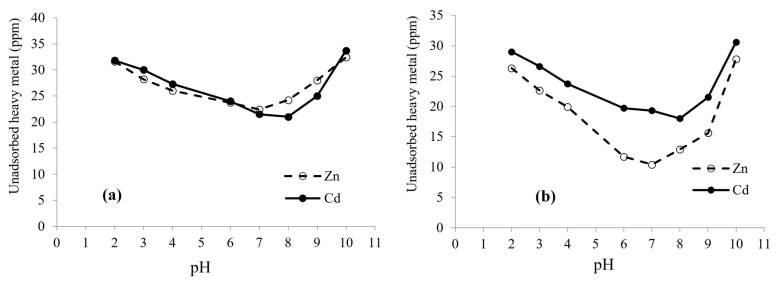
Effect of pH on biosorption of Zn^2+^ and Cd^2+^ by living (**a**) and lyophilized (**b**) biomass of *Priestia megaterium* PRJNA526404.

**Figure 4 microorganisms-10-02510-f004:**
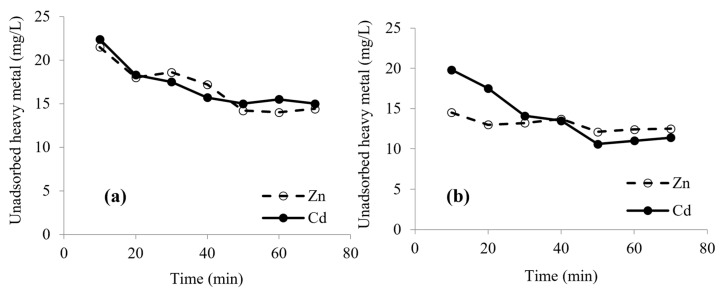
Effect of time on biosorption of Zn^2+^ and Cd^2+^ by living (**a**) and lyophilized (**b**) cells of *Priestia megaterium* PRJNA526404.

**Figure 5 microorganisms-10-02510-f005:**
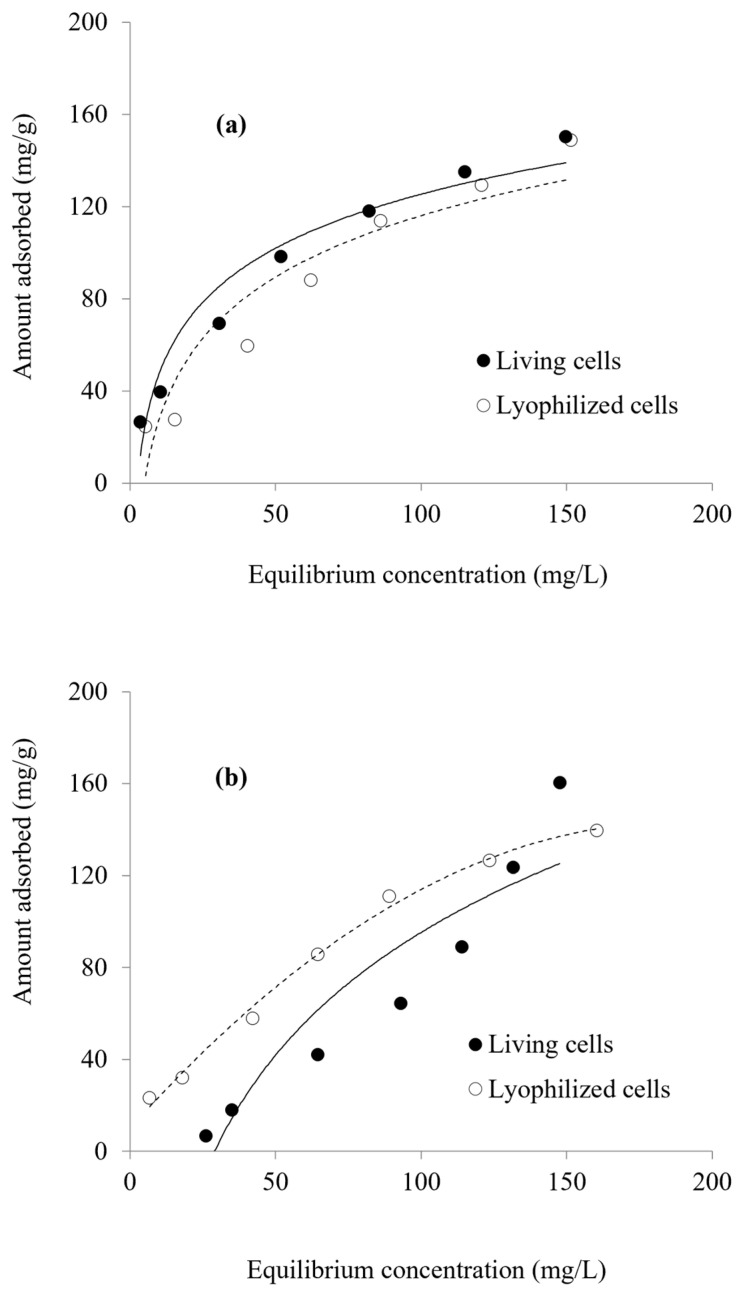
Zn^2+^ (**a**) and Cd^2+^ (**b**) biosorption by living and lyophilized biomass of *Priestia megaterium* PRJNA526404.

**Figure 6 microorganisms-10-02510-f006:**
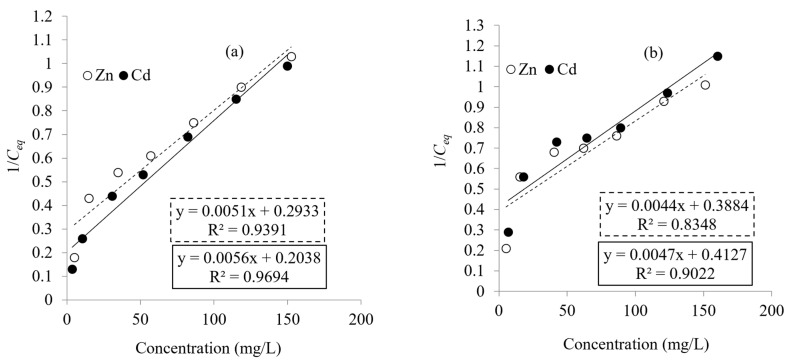
Linear form of Langmuir adsorption isotherms for adsorption of Zn^2+^ and Cd^2+^ by living (**a**) and lyophilized (**b**) cells of *Priestia megaterium* PRJNA526404.

**Figure 7 microorganisms-10-02510-f007:**
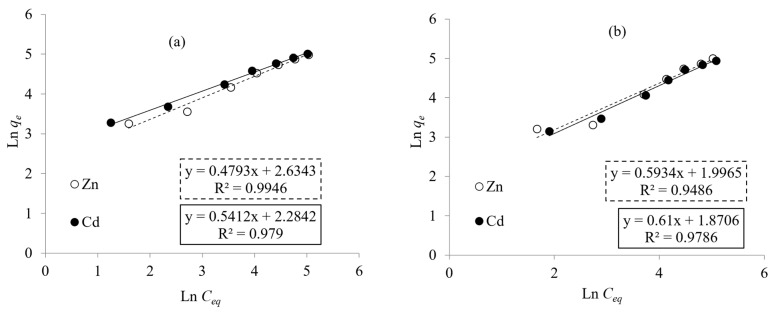
Linear form of Freundlich adsorption isotherms for adsorption of Zn^2+^ and Cd^2+^ by living (**a**) and lyophilized (**b**) cells of *Priestia megaterium* PRJNA526404.

**Figure 8 microorganisms-10-02510-f008:**
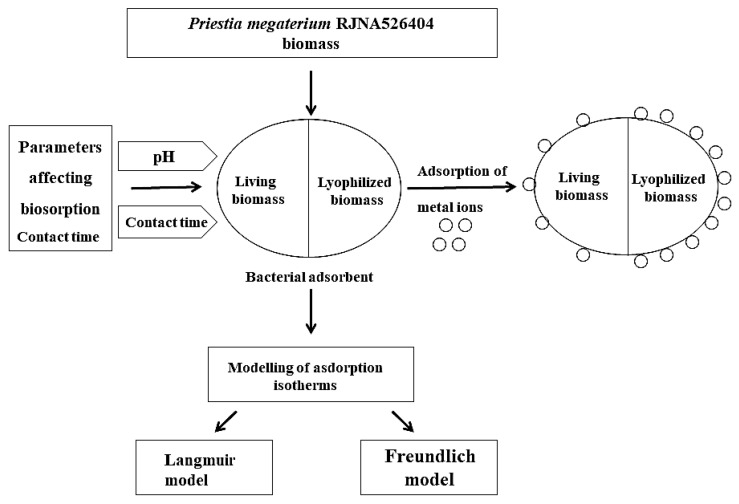
Schematic illustration of *Priestia megaterium* PRJNA526404 biomass preparation and adsorption procedure.

**Table 1 microorganisms-10-02510-t001:** Morphological and biochemical characteristics of strain *Priestia megaterium* PRJNA526404.

Characteristics Observed	Tests Employed
+ve	Gram stain
bacilli	Morphology
pairs, chains	Arrangement
+ve	Endospore
+ve	Starch hydrolysis
+ve	Catalase
−ve	Oxidase
+ve	Acid from glucose
−ve	VP
+ve	Citrate
+ve	Urease
−ve	Nitrate reduction
+ve	Glucose
+ve	Lipase production
+ve	Growth on 7% NaCl
Obligate aerobe	O/F

“+” and “−” indicate positive and negative reactions, respectively.

**Table 2 microorganisms-10-02510-t002:** Langmuir and Freundlich adsorption constants calculated from the Langmuir adsorption isotherms of Zn^2+^ and Cd^2+^ by living and lyophilized biomass of *Priestia megaterium* PRJNA526404.

	Living Cells	Lyophilized Cells
	Zn^2+^ Cd^2+^	Zn^2+^ Cd^2+^
**Langmuir** *q_max_* (mg/g)	196.08 178.57	227.27 212.777
b (L/mg)	0.017 0.027	0.0113 0.0114
r^2^	0.9391 0.9694	0.8348 0.9022
**Freundlich** K_f_ (mg/g)	7.36 6.49	13.93 9.82
n (L/mg)	2.09 1.85	1.69 1.64
r^2^	0.9946 0.979	0.9486 0.9786

**Table 3 microorganisms-10-02510-t003:** Maximum biosorption capacity values (*q_max_*) obtained by other biosorbents in comparison with (*q_max_*) values of *Priestia megaterium* PRJNA526404.

Metal Ion	Biosorbent	*q_max_* (mg/g)	Reference
**Zn^2+^**	*Priestia megaterium* PRJNA526404	227.27	The present work
	*Saccharomyces cerevisiae*	3.462	[9]
	*Caulerpa lentillif*	40.7	[53]
	*P. aeruginosa* B237	16.75	[54]
	*Morganella morganii* ACZ05	31.24	[55]
	*Citrobacter Strain* MCM B-181	26.38	[27]
**Cd^2+^**	*Priestia megaterium* PRJNA526404	212.777	The present work
	*Saccharomyces cerevisiae*	4.73	[9]
	*P. aeruginosa* B237	16.89	[54]
	*Exiguobacterium* sp.	15.6	[55]
	*Citrobacter Strain* MCM B-181	66.10	[27]

## Data Availability

Not applicable.

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
