# Peer review of "Improvement of Zn (II) and Cd (II) Biosorption by *Priestia megaterium* PRJNA526404 Isolated from Agricultural Waste Water"

_microorganisms, 2022, doi:10.3390/microorganisms10122510_

Round 1

Reviewer 1 Report

Detailed comments:

1.      The English of the text should be checked

2.      The novelty of the manuscript is complete missing

3.      In the Introduction part must be included more information about other materials or methods used for removal of heavy metals from water or wastewater. The following references can be included in the Introduction part to improve the quality of manuscript, because they provide relevant information:

ü  Adsorption-oriented processes using conventional and non-conventional adsorbents for wastewater treatment”. Green Adsorbents for Pollutant Removal, 18, Springer Nature, 2018, 23-71, Environmental Chemistry for a Sustainable World, 978-3-319-92111-2.

ü  Oxide Layered Double Hydroxide Materials: Synthesis, Characterization and Efficient Application for Mn2+ Removal from Synthetic Wastewater, Materials, 2020, 13(18), 4089

ü  Low-cost Adsorbents Derived from Agricultural By-products/Wastes for Enhancing Contaminant Uptakes from Wastewater: A Review”, Pol. J. Environ. Stud. 26(2), 2017, 479-510

ü  Green Adsorbents for Wastewaters: A Critical Review”, Materials 7, 2014, 333-364.

ü  Removal of heavy metals from emerging cellulosic low-cost adsorbents: a review, Appl. Water Sci. 7(5), 2016, 2113-2136.

ü  Removal of nickel ions from synthetically wastewater using membrane doped with natural  biosorbent, Romanian Journal of Physics, 64 (9-10), 2019, 821

4.      For all parameters used in the equations indicate what represent and unit of measure

5.      For all abbreviations or notation indicate the complete name

6.      For unit of measure used the S.I., please check in all manuscript.

7.      All operational conditions must be indicated, amount, concentration, time, speed rotation

8.      For all reagents or chemicals used must be indicated manufacturer, purity, concentration, amount

9.      All equipment and tools used in this study should be described in detail or further information should be provided (manufacturer, type, operational conditions, etc)

10.  The value for room temperature must be indicated

11.  At figures 4, 5, 6, 10 – please indicate at axes: parameters, scale, unit of measure, legends

12.  At 2.2 Preparation of Priestia megaterium PRJNA526404 biomass and 2.7. Adsorption procedure – schematically diagrams or photos/pictures must be included/indicated

13.  At all figures, at axes indicate unit of measure or correct the legends

14.  For heavy metals, indicate ionic valence, in all manuscript, tables, figures, legends (e.g., replace Zn with Zn2+)

15.  Comparison between the obtained results and measured in this study with other reported studies should be done and included for more clarity (indicate values not just number of reference)

16.  More Conclusions with the best obtained results.

17.  The other possible applications for materials used in this manuscript must be related at the final of the manuscript or at Conclusions

Author Response

Reviewer 1 Response

  1. The English of the text should be checked.

R: The English of the text was checked.

  1. The novelty of the manuscript is complete missing

R: the novelty of the manuscript is approximately found whereas the data obtained  in the present study was unique and the max absorption was remarkably high compared to other published reports. 

  1. In the Introduction part must be included more information about other
    materials or methods used for removal of heavy metals from water or
             wastewater. The following references can be included in the Introduction part
             to improve the quality of manuscript, because they provide relevant  
             information:

ü  Adsorption-oriented processes using conventional and non-conventional adsorbents for wastewater treatment”. Green Adsorbents for Pollutant Removal, 18, Springer Nature, 2018, 23-71, Environmental Chemistry for a Sustainable World, 978-3-319-92111-2.

ü  Oxide Layered Double Hydroxide Materials: Synthesis, Characterization and Efficient Application for Mn2+ Removal from Synthetic Wastewater, Materials, 2020, 13(18), 4089

ü  Low-cost Adsorbents Derived from Agricultural By-products/Wastes for Enhancing Contaminant Uptakes from Wastewater: A Review”, Pol. J. Environ. Stud. 26(2), 2017, 479-510

ü  Green Adsorbents for Wastewaters: A Critical Review”, Materials 7, 2014, 333-364.

ü  Removal of heavy metals from emerging cellulosic low-cost adsorbents: a review, Appl. Water Sci. 7(5), 2016, 2113-2136.

ü  Removal of nickel ions from synthetically wastewater using membrane doped with natural  biosorbent, Romanian Journal of Physics, 64 (9-10), 2019, 821

R: The above  references were included in the Introduction.

  1. For all parameters used in the equations indicate what represent and unit of
             measure

R: All parameters used in the equations were indicated.

  1. For all abbreviations or notation indicate the complete name.

R: Complete names for all abbreviation have been noted.

  1. For unit of measure used the  please check in all manuscript.

R: Units of measure used the   was checked in all manuscript.

  1. All operational conditions must be indicated, amount, concentration, time,
    speed rotation.

R: All operational conditions, amount, concentration, time, speed rotation  were indicated.

  1. For all reagents or chemicals used must be indicated manufacturer, purity,
    concentration, amount

R: For all reagents or chemicals used were indicated manufacturer.

  1. All equipment and tools used in this study should be described in detail or
    further information should be provided (manufacturer, type, operational
            conditions, etc)

R: All equipment and tools used were manufacturer indicated.

  1. The value for room temperature must be indicated

R: The valued for room temperature was indicated.

  1. At figures 4, 5, 6, 10 – please indicate at axes: parameters, scale, unit of
    measure, legendsk.

R: All  were indicated.

  1. At 2.2 Preparation of Priestia megaterium PRJNA526404 biomassand
           2.7. Adsorption procedure – schematically diagrams or photos/pictures must  
           be included/indicated

R: Schematically diagrams was included as Figure 8.

  1. At all figures, at axes indicate unit of measure or correct the legends

R: It was done.

  1. For heavy metals, indicate ionic valence, in all manuscript, tables, figures,
    legends (e.g., replace Zn with Zn2+)

R:  it was done.

  1. Comparison between the obtained results and measured in this study with
           other reported studies should be done and included for more clarity (indicate
           values not just number of reference).

R: The comparison was included as a Table 3.

  1. More Conclusions with the best obtained results.

R: It was done.

  1. The other possible applications for materials used in this manuscript must be
    related at the final of the manuscript or at Conclusions.

R: It was done.

Reviewer 2 Report

In this manuscript mentioned the zinc and cadmium adsorption properties of both frozen cells and microorganisms .As for the research approach method, we have selected a typical method for adsorption research, and we think that there is no problem with the results and their arrangement. However there  are some points that should be corrected, so please revise.

L41 underline

Figure 2 ,3 4,6,7: Figure tile put on under figure5.

I can not find Figure 5.

L280 table(2)→ Table 2

Figure 6 left side : location of Ceq

Figure 7 :lnqe→ ln qe

Do you need  create regression equation with initial and equilibrium concentration ?

Fonts different : Reference 5,

Line 449 ,460  no title, authors name ,no journal name 

underline: Line 378,,388,420,,440,467

Figure 6: 

Author Response

Reviewer 2 Response

In this manuscript mentioned the zinc and cadmium adsorption properties of both frozen cells and microorganisms .As for the research approach method, we have selected a typical method for adsorption research, and we think that there is no problem with the results and their arrangement. However there  are some points that should be corrected, so please revise.

L41 underline

R: It was deleted.

Figure 2 ,3 4,6,7: Figure tile put on under figure5.   ?

R: It was done.

I can not find Figure 5.

R: It was added.

L280 table(2)→ Table 2

R: It was done.

Figure 6 left side : location of Ceq

R: It was done.

Figure 7 :lnqe→ ln qe

R: It was done.

Do you need create regression equation with initial and equilibrium concentration ?

R: It was added as Figure 5.

Fonts different: Reference 5,

R: It was corrected.

Line 449 ,460  no title, authors name ,no journal name

R: It was done. 

underline: Line 378,,388,420,,440,467

R: It was done. 

Round 2

Reviewer 1 Report

Accept

Correct, at legends (at Figures), Zn with Zn2+, and Cd with Cd2+